

# Comparing efficacy in reducing pulling and welfare impacts of four types of leash walking equipment

Anamarie C. Johnson and Clive D. L. Wynne

Department of Psychology, Arizona State University, Tempe, Arizona, United States

## ABSTRACT

**Background:** Leash pulling is a commonly reported problem behavior for dog owners, as a result, a variety of leash equipment types are offered to mitigate pulling force. We were particularly interested in prong collars as their inherently aversive design has made their use a subject of debate. Though banned in certain countries and widely available in others, to date, there is no research comparing them to other leash walking equipment.

**Methods:** We compared four types of leash walking equipment: a martingale (flat collar as baseline measure), a front-connection harness, a polymer prong-style collar (Starmark), and a standard metal prong collar. Twenty-three dogs were walked on all four types of equipment for 5-min each. Equipment was attached to a leash which was connected to a battery-powered strain gauge to measure the dog's pulling force. All walks were video recorded for behavior analysis.

**Results:** There were statistically significant differences among the leash equipment types in pulling impulse (Newtons × seconds), ($\chi^2_{(2)}$ = 30.6, $p$ < 0.001). *Post-hoc* analysis revealed significant differences in impulse between the martingale and the other equipment: harness (Z = −3.69, $p$ < 0.001), Starmark collar (Z = −3.62, $p$ < 0.001) and prong collar (Z = −3.92, $p$ < 0.001). No other differences among equipment types were significant. Fifteen behaviors were examined as welfare indicators but only three: looking at the handler, lip licking, and sniffing occurred across all dogs and all walks. There was a statistically significant difference in frequency of lip licking behavior across the four types of leash-equipment ($\chi^2_{(2)}$ = 8.17, $p$ = 0.04) and *post-hoc* analysis showed a difference between the martingale and the harness (Z = −2.65, $p$ = 0.008). While our research did not provide any clear evidence of poorer welfare due to equipment type, we caution the generalizability of these findings and recommend further assessment of these items of leash-walking equipment in real-life scenarios.

Corresponding author
Anamarie C. Johnson,
acjohn47@asu.edu

## INTRODUCTION

There are an estimated 65 million pet dogs in the United States with an additional 113 million in Europe (*HealthforAnimals: Global Animal Health Association, 2022*). *Blackwell et al. (2008)*, reported that 69% of United Kingdom dog owners felt leash pulling to be a problem. A more recent survey of owners in the U.K. and Ireland found that 82.7%

reported their dog pulled on leash (*Townsend et al., 2021*). Over thirty days of study, 32% of dogs that pulled on leash received less than a 30 min walk each day and a further 18% were not walked every day (*Townsend et al., 2021*). Similar trends were observed in another U.K. sample in which 9% of respondents reported that they had concerns about their dog's behavior on leash and, as a result, 10% reported that they walked the dog less than 30 min per day (*The People's Dispensary for Sick Animals (PDSA), 2020*). Overall, an owner's concern about managing their dog on leash leads to dogs not receiving daily exercise, which can lead to weight gain and other behavioral concerns from limited socialization and enrichment (*Townsend, Dixon & Buckley, 2022*).

Owners report using a variety of leash-walking equipment to try and manage their dog's behavior. Flat collars are the most common piece of equipment for both U.S. and U.K. dog owners while harnesses, both back-connection and front-connection, follow in popularity (*Dinwoodie, Zottola & Dodman, 2021a*; *Townsend et al., 2021*). *Dinwoodie, Zottola & Dodman (2021a)* reported that 28% of owners of dogs under the age of 6 months (pre-adolescence) who attended reward-based training classes still reported a use of aversive devices like prong collars (6% of all respondents), slip collars (9%), and choke chains (5%). In a study of dog owners surveyed at a shelter, those who walked their dogs on a prong collar or choke chain reported being less satisfied with their dog's overall and leash-walking behavior regardless of whether they were relinquishing their dog to the shelter or continuing with ownership (*Kwan & Bain, 2013*). In a survey of owners with dogs presenting at least one type of aggressive behavior, owners of dogs with fear aggression to dogs appreciated the immediate control they achieved with aversive collars like slip, choke, and prong collars. However, as the authors noted, these punishment-based methods can have long-term negative effects (*Dinwoodie, Zottola & Dodman, 2021b*).

Given the wide range of equipment that owners use to tackle leash pulling, several studies have tried to assess the potential physiological and behavioral effects of different leash-walking equipment and how they might impact welfare. *Carter, McNally & Roshier (2020)* compared the force and pressure of seven commercially available collars on a simulated canine neck model. With each collar on the model and attached to a leash, three different force intensities were administered: a 40 Newton (N) light constant pressure, a 70 N strong constant pressure, and a sharp jerk with a pressure around 141 N (*Carter, McNally & Roshier, 2020*). The researchers reported that all recorded pressures across all collars and all magnitudes exerted a level of pressure known to cause tissue damage and death in humans; however, the use of a simulated neck model did not provide insight into how the model might differ from dog physiology. Overall, they noted that focusing on what type of collar is best is not important if any collar at the lowest magnitude of pulling has the potential to cause injury (*Carter, McNally & Roshier, 2020*). Another study measured the resulting intraocular pressure created from a dog pulling on a neck collar or a harness (*Pauli et al., 2006*). Intraocular pressure significantly increased in dogs when pulling on the neck collar but not on the harness, leading the authors to recommend dogs with or at risk of eye disease to wear harnesses rather than collars while exercising (*Pauli et al., 2006*).

A comparison study of a neck collar and a head collar (in which dogs are controlled by their snout), found no significant differences in two common measures of physiological stress —plasma adrenocorticotropic hormone (ACTH) or cortisol (*Ogburn et al., 1998*). Similarly, there were no significant differences in the number of vocalizations and occurrences dragging behind the handler. When wearing the neck collar during obedience exercises, dogs looked to the handler and had their ears back significantly more often than with the head collar but crouched more when wearing the head collar. Overall, when wearing the head collar, dogs pawed at the device on their nose and were described as "more unruly". In another study comparing a neck collar to a harness, dogs with a history of wearing a neck collar held their ears back while wearing either the collar or the harness (*Grainger, Wills & Montrose, 2016*). However, the authors cautioned that due to the lack of other stress indicators, the observation of ears back should be taken with caution. The authors noted that the lack of behavioral stress responses indicated that welfare was not compromised on either device (*Grainger, Wills & Montrose, 2016*).

Fewer studies have investigated how dog pulling can be measured and may vary when dogs are walked on different equipment types. A study by *Shih et al. (2020)* was the first to walk dogs on a strain gauge to measure leash tension. Dogs were walked on a front-connection harness with the leash attached to both the front attachment and a neck collar for safety. Researchers were able to demonstrate that there was a significant effect of dog size and body weight on leash tension and that, overall, the strain gauge was a valid approach to investigate real-life walking interactions.

A follow up study investigated the differences in pulling between a neck collar and a back-connection harness: researchers were interested in exploring anecdotal information that dogs pull more on a back-connection harness (*Shih et al., 2021*). In this study, shelter dogs wearing either a neck collar or harness were attached on a leash to the strain gauge fixed to a wall. Dogs were encouraged to pull by either presenting a treat or a squeaky ball 50 cm away. Pulling time was defined as the time dogs pulled on leash with a force greater than 1% of the dog's body weight. Dogs pulled with greater force and for longer while wearing the harness than the collar. Investigators theorized that dogs might have pulled more on the harness than the collar because it was more comfortable. As in the *Grainger, Wills & Montrose (2016)* study, there were no significant differences in common behavioral stress indicators such as lip licking, panting, or tail position; dogs wearing the harness did spend more time looking at the handler in the food condition, but the authors hypothesized that this could be because the harness made it more comfortable to turn and look compared to the collar (*Shih et al., 2021*).

Despite the widespread use of prong collars (*Dinwoodie, Zottola & Dodman, 2021a*; *Kwan & Bain, 2013*), little research has investigated their use, particularly in relation to their ability to mitigate leash pulling. The prong collar is not a new invention with versions comparable to the modern device dating back to the 1800s as a method to punish or correct a dog, particularly birding dogs (*Carleton, 1849*). A United States patent from 1878 depicts a leather collar with three rotating spikes which is noted as an "improvement in force-collars for breaking dogs" (*Von Culin, 1878*, p. 1). A book on training retrieving

dogs referred to a similar device as a "spike collar" depicting a collar with a leather strap with internal spikes (*Waters, 1895*). Modern patents on prong collars and prong-style devices note that such collars are important for "animal control, training, and behavior modification" (*Wolfe et al., 2003*, p. 6).

The only study to date that has specifically examined the use of a prong collar compared to an electronic shock collar (e-collar), prong collar, and conditioned quitting signal across three training sessions in Belgian Malinois police dogs (*Salgirli, Schalke & Hackbarth, 2012*). In a within-subject design, each dog experienced the three training conditions in which researchers assessed the effectiveness of the training method to correct a dog with positive punishment (adding an aversive stimulus to reduce the likelihood of a behavior occurring again) after breaking a "heel" command and going towards a decoy handler. If the dog successfully stopped at the correction, the same procedure was later tested to assess learning. Welfare was assessed through salivary cortisol measures and behavioral analysis. While not statistically significantly different, 38% of dogs on the e-collar presented a backward ears position, compared to 64% on the prong collar. Nearly 5% of dogs on the prong collar presented crouching body posture; crouching was not observed in the e-collar condition. When comparing maximum cortisol values to resting cortisol, cortisol values after the prong and the e-collar were lower than basal values (*Salgirli, Schalke & Hackbarth, 2012*).

Although the Salgirli study is the only study to focus on the efficacy and welfare effects of the prong collar, it was used in a very specific context, albeit not fully described, with a unique population of dogs. The more common use of a prong collar is to reduce leash pulling (*Dinwoodie, Zottola & Dodman, 2021a*; *Kwan & Bain, 2013*) using either a jerk on the leash when the dog pulls forward, (positive punishment), or alleviation of prong pressure on the neck when it stops pulling (negative reinforcement). Several organizations do not support the use of prong collars because they are aversive methods (*American Veterinary Society of Animal Behavior (AVSAB), 2021*; *RSPCA, 2024*), but there is no research to determine whether they are any better or worse at reducing pulling and whether their use induces negative welfare. As a result, the aim of this study was to evaluate how effective the prong collar and three other types of commercially available leash walking equipment were in reducing pulling and if they varied in stress-related behaviors. Dogs were walked on a strain gauge to assess differences in pulling force across the devices and sessions were video recorded to assess differences in behavior. Portions of this manuscript were previously published as part of a preprint (*Johnson, 2024*).

## MATERIALS AND METHODS

This study was approved by the Institutional Animal Care and Use Committee (IACUC) of Arizona State University (Protocol Number: 22-1934R).

### Participants and location

*A priori* power analysis for an ANOVA, with repeated measures, with a power of 0.8 and medium effect size of 0.25 indicated a sample size of 24 was required.

**Table 1 Participating dogs' age, breed and weight.**

| Name | Age estimate at time of study (months) | Shelter breed estimate | Weight (in kilograms) at time of study | Shelter |
|---|---|---|---|---|
| Dominic | 60 | German shepherd | 34.02 | 1 |
| Duke | 84 | Rottweiler/German shepherd mix | 30.26 | 1 |
| Harley | 36 | Doberman mix | 28.71 | 1 |
| Mr. Fitz | 108 | Pit bull mix | 19.50 | 1 |
| Ranger | 12 | Australian cattle dog mix | 23.32 | 1 |
| Royal | 60 | Pit bull mix | 31.12 | 1 |
| Sunny | 20 | German shepherd/husky mix | 15.12 | 1 |
| Bubba | 19 | Pit bull mix | 26.68 | 1 |
| Daisy | 72 | German short haired pointer/pit bull mix | 25.22 | 1 |
| Mars | 60 | Pit bull mix | 26.17 | 1 |
| Sulley | 12 | Mastiff mix | 26.26 | 1 |
| Buzz | 10 | Husky mix | 17.92 | 1 |
| Jake | 10 | German shepherd | 24.04 | 1 |
| Saint | 24 | German shepherd | 33.57 | 1 |
| Juju | 84 | Husky mix | 26.76 | 1 |
| KimChi | 36 | Siberian husky/German shepherd mix | 20.87 | 2 |
| Mrs. Potato head | 60 | Pit bull mix | 28.58 | 2 |
| Reggie | 60 | Pit bull mix | 37.65 | 2 |
| Richa | 120 | Presa canario/giant schnauzer | 43.09 | 2 |
| Ursula | 13 | Pit bull mix | 20.41 | 2 |
| Boris | 24 | Doberman | 41.50 | 2 |
| Klutz | 12 | Belgian malinois mix | 33.57 | 2 |
| Tyr | 48 | German shepherd | 42.64 | 2 |

We identified 26 dogs from two Sacramento, California area shelters, 23 of whom completed all four walks (Table 1). The remaining three dogs were excluded from the study because they were too excitable, jumping and mouthing on the handler while attempting to fit the martingale, and considered unsafe. All care, feeding, and enrichment was guided and provided by shelter staff and volunteers. All decisions regarding outcomes for these animals—whether placement in an adopted or foster home, medically- or behaviorally-necessary humane euthanasia, or other—was also determined by the shelter staff. This study had no impact on any of these factors. We chose to work with shelter dogs to have a population of study, dogs with consistent recent exposure to just one form of leash walking equipment (a slip lead) and consistent recent walking experience including, at a minimum, once daily opportunities with volunteers in outdoor exercise yards.

Because of the available size of leash-walking equipment, we selected only medium to large-sized (13–42 kg) dogs between the ages of 6 months and 10 years old to participate. To participate in the study dogs had to be available for adoption, implying that they had passed a shelter behavior assessment, and had to be willing to go for walks led by a novel person. Shelter staff indicated which dogs fulfilled these criteria. Given that multiple dogs

were eligible at each shelter, the first author selected from among eligible dogs by random number assignment.

The study took place over four days: September 12 and 13, 2022 at Shelter 1 and September 14 and 15 at Shelter 2. Both shelters are public, municipal shelters receiving stray and owner-relinquished animals from their communities. Shelter 1 was built in the 1970's with dogs living in kennel wards of concrete construction and wire fence fronts. Shelter 2 opened in 2019. Dogs were also housed in kennel wards but with plastic, clear fronts. Both shelters had shelter staff overseeing daily cleaning of the wards and morning and evening feedings for the dogs. Additionally, both shelters had robust volunteer programs where volunteers will take dogs out to outdoor yards to socialize and play. If considered appropriate, dogs were also provided the opportunity for play with another dog. Dogs at both shelters were walked by volunteers and staff to different kennel locations or areas within the shelter like the outdoor yards on slip-leads.

## Equipment

Dogs were walked on four types of leash walking equipment: 1-inch-wide (2.54 cm) martingale collar (Radio Systems Corporation, Knoxville, TN, USA), front-connection harness (2 Hounds Design, Indian Trail, NC, USA), Starmark collar (Starmark Pet Products, Hutto, TX, USA) and prong collar (Herm Sprenger USA, Queensbury, NY, USA). Each type of equipment was then attached to a five-foot-long, (1.52 m) 1-inch-wide (2.54 cm) nylon leash. Dogs were walked between sessions on a slip-lead (Mendota, Saint Paul, MN, USA).

A martingale collar (Fig. 1A) is made of nylon fabric; a leash is attached to a connector on the loop portion of the collar. A martingale is designed to tighten as the dog pulls and is often used as a safety mechanism compared to a regular dog collar which a dog may slip through during a walk. A front-connection harness (Fig. 1B) is designed with a leash attachment point at the dog's chest and is often advertised as "no-pull" compared to back-connection harnesses which anecdotally encourage pulling (Shih et al., 2021). Both a Starmark collar (Fig. 1C) and a metal prong collar (Fig. 1D) are designed like a martingale collar with a leash attached to the loop portion of the collar—the collars tighten as the dog pulls and loosen as the dog stops pulling. A Starmark collar is designed, like a prong collar, to "provide pressure to the animal so as to assist in controlling and training the animal" (Wolfe et al., 2003, p. 1). However, a Starmark collar has polymer V-shaped projections to apply this pressure, unlike a prong collar that utilizes metal links. As a result, the Starmark collar was designed to be a middle ground between collars without any aversive elements to highly aversive collars with metal prong projections (Wolfe et al., 2003).

To measure the pulling force of each dog in each condition, we created a battery-powered strain gauge (Fig. 2) that dynamically measured pulling force over the course of a walk. This gauge was attached to the nylon leash which was then attached to each type of equipment.

The strain gauge was constructed from three microcomputers (Adafruit, New York, NY, USA) and a load cell amplifier connected to a high-precision load cell scale sensor able to measure up to 100 kg (981 N). This system was mounted to an ascender (NewDoar, Las

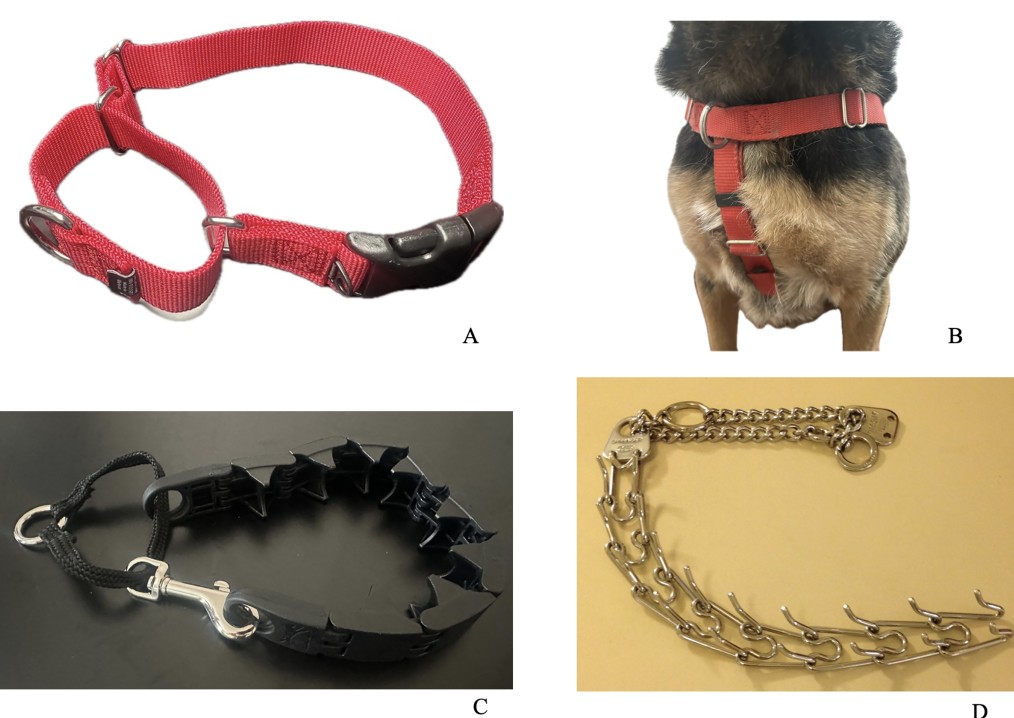

**Figure 1 Images of leash-walking equipment used in the study.** (A) Martingale collar, photo credit: own photo. (B) Front-connection harness, photo credit: own photo. (C) Starmark collar, photo credit: own photo. (D) Prong collar, photo credit: polymath38 *via* Wikimedia Commons.

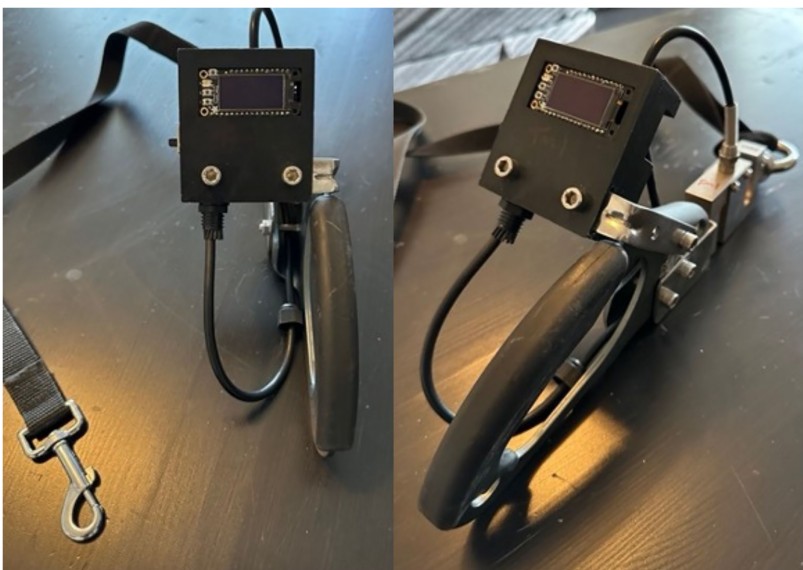

**Figure 2 Strain gauge.** The panel on the left depicts the computer attached to the ascender handle held by handler. The panel on the right depicts eye-bolt attachment connected to the load-cell sensor on one end and leash on the other. Source: own photo.

Vegas, NV, USA) designed for rock climbing and utilized in this study as a large handle. An eyebolt (Muzata, Compton, CA, USA) was then welded onto the ascender and load cell sensor and the nylon leash was then attached to the eyebolt.

The strain gauge was calibrated by attaching various weights the assess measurement accuracy. Additionally, before it was used at either shelter, several dogs were walked with the strain gauge to determine that data was being accurately recorded during a walk and could be appropriately exported.

All walks were recorded using a Insta360 One X2 camera (Insta360, Irvine, CA, USA). The camera has one wide angle lens on each side of the device that can record in 360 degrees and can then be edited later to keep the focal subject within frame of the resulting video. The camera was held on an extendable stick by the handler's arm parallel to the dog; this allowed the dog's entire body to be recorded during the walk.

## Procedure

Each dog was walked on the slip lead from its home kennel to an enclosed outdoor yard. The handler allowed the dog to explore the outdoor yard off-leash for several minutes before the dog was fitted with one of the leash equipment types. This equipment was then attached to the nylon leash attached to the strain gauge.

It was important to ensure the dogs were maximally energetic throughout the measured walks, so after piloting of walks 10 min in duration, all walks from which data were collected lasted 5 min. All dogs were walked on each piece of equipment for 5-min beginning as soon as the handler and dog exited the outdoor yard area. Video recording and tension measurement also began at this time. The first author was the handler for all dogs and walks.

All walks were completed in the same order of increasing potential level of aversiveness (martingale collar, front-connection harness, Starmark collar, prong collar) out of a concern that tools designed to provide aversive stimulation could cause discomfort to a dog and influence its behavior with the same handler on future walks. Potential aversiveness was interpreted in this study as the device's potential to impose discomfort on the dog given each tool's design and construction. The martingale and harness are both made from fabric that lies flat on the dog's body. However, the harness requires additional handling to affix to the dog and has multiple pressure points on the body which can make it more restrictive than a simple collar (*Lafuente, Provis & Schmalz, 2019*; *Pálya et al., 2022*). Both the Starmark and prong collars have projections designed to create pressure and tension on a dog's neck to stop the dog from pulling (*Wolfe et al., 2003*). The projections on the prong collar are stainless steel, which is likely more aversive than the plastic prongs on the Starmark collar.

At the end of each walk, the dog was brought back to the enclosed outdoor yard where the equipment was taken off, and then walked back on the slip lead to its home kennel. Dogs were provided at least 2 h in their home kennel between walks.

The handler walked the dog as neutrally as possible and kept quiet throughout the walk. To ensure all pulling force on the leash was guided by the dog and not imposed by the handler, the handler followed the dog wherever it chose to walk. If the dog chose to stop,

the handler remained still for up to 1 min before attempting to engage the dog to continue the walk.

The handler only controlled the direction and path of the dog if necessary for safety. The dog was not allowed to walk off the curb when walking on a sidewalk; if it attempted to walk off the curb, the handler stood still and waited until the dog resumed walking along the sidewalk. The handler also remained still if the dog pulled towards an object that might be dangerous or inedible for the dog. At any point, if the dog was continually exerting pulling force on the leash towards a specific object or in a specific direction that it could not reach, the handler allowed the dog to engage in pulling for 1 min before attempting to engage with the dog and coax it in an alternative direction.

Behavior was continually assessed during the walks by the handler to identify any acute detrimental impacts. These acute impacts could be vocalizations like yelping or howling, refusal to walk on the tool, such as opposing forward movement or lying on the ground. Given the nature of these tools, any acute pain could be immediately alleviated by stopping the walk and, if necessary, replacing the leash-walking tool with a slip lead.

Data collection (measured tension and video) was terminated after 5 min whether or not the dog and handler had returned to the starting location.

## Video analysis

A total of 92 videos, four per dog, was analyzed using the event logging software BORIS (*Friard & Gamba, 2016*), operated by video coders trained to 80% or better accuracy. Video coders were blind to the aims of the study. A random 20% of the videos was double coded to ensure reliability.

Behaviors were selected for the ethogram due to their use in prior studies to assess stress-related behaviors (*e.g.*, *Deldalle & Gaunet, 2014*; *Grainger, Wills & Montrose, 2016*; *Petrean et al., 2023*; *Shih et al., 2021*). However, we were unable to code for flattened or lowered ears, commonly reported as a signal of stress (*e.g.*, *Grainger, Wills & Montrose, 2016*), because of the difficulty in determining ear orientation during a walk. All behaviors were coded for frequency (Table 2).

## Data and statistical analysis

The microcomputers in the strain gauge were designed to record force pulling every quarter second which was then exported for analysis. The quarter-second data were cumulated across the 300 s of each walk and the total impulse (total force (Newtons) × seconds) for each dog on each type of leash walking equipment was calculated.

All statistical analyses were conducted using the software package SPSS (Version 28; IBM, Chicago, IL, USA).

All data were tested for normality with Shapiro-Wilk's test. Because significant departures from normality were observed, non-parametric tests were deployed. Friedman's test was conducted on the impulse data for the four types of leash walking equipment, as well as the frequency of the three most commonly occurring behaviors. Wilcoxon signed ranks test with Bonferroni corrections were completed to determine significant differences

**Table 2 Ethogram of fifteen behaviors coded in video behavior analysis.**

| Coded behavior | Operational definition |
|---|---|
| Whale eye | Dog's eyes are wide, whites may be visible. |
| Looking at handler | Dog turns towards and looks at handler (face, body, hands). |
| Barking | Dog barks. |
| Yelping | Quick, sharp vocalization, may be once or over a few seconds. |
| Whining | Dog gives high pitched whine, may be quick or over a prolonged period of time. |
| Howling | Dog howls, mouth may be open slightly or fully. |
| Growling | Dog gives a low, deep growl. |
| Lip licking | Dog moves part of tongue out of mouth and drags it along upper lip. |
| Lowered tail | Dog's tail is held down straight or tucked between leg. |
| Crouch | Dog may be standing upright then crouch down. Shoulders and hindquarters will follow, giving dog a "rounded" appearance. Head may be lower than torso. |
| Yawning | Dog opens mouth wide and yawns, may occur with or without vocalization. |
| Shake off | Dog shakes body and/or head, like how a dog may shake off water after a bath. |
| Quivering | Dog is trembling/quivering. |
| Sniffing | Dog directs nose downward or upward to sniff an item or substrate for longer than 1 second, end of sniffing bout signified by dog lifting head which can be accompanied by walking away from the original focal object. |
| Balk | Dog resists moving/walking forward with handler, dog may sit or move backwards to avoid forward movement. |

between equipment types. An alpha level of 0.05 was adopted throughout (corrected as necessary for multiple comparisons).

# RESULTS

Figure 3 shows total impulse (Ns) over the 5-min walks for the four equipment types. Impulse differed significantly among the four types of leash-walking equipment (Friedman's test, $\chi^2_{(2)} = 30.6$, $p < 0.001$). *Post-hoc* analysis with Wilcoxon signed-ranked tests using a Bonferroni corrected significance level of $p$-level 0.008 revealed significant differences in impulse between the martingale and all the other equipment types: harness ($Z = -3.69$, $p = 0.001$, $r = 0.77$), Starmark collar ($Z = -3.62$, $p = 0.001$, $r = 0.75$) and prong collar ($Z = -3.92$, $p = 0.001$, $r = 0.82$). No other differences in impulse among forms of equipment were significant (Fig. 2).

All walks for all dogs were fully recorded and analyzed. Of the 15 behaviors analyzed from the ethogram (Table 2), only three: looking at handler, lip licking, and sniffing, occurred with sufficient frequency for statistical comparisons. To confirm coding reliability for these specific behaviors, percent agreement was calculated- 90.7% agreement for sniffing, 91.1% for looking at handler, and 84.4% for lip licking. The remaining behaviors occurred less than 10 times per dog (Table 3). Vocalizations were very infrequent across any of the dogs; if they did occur, they were in response to external stimuli like people or other dogs, not in obvious relation to distress from the leash equipment type. Additionally, resisting forward movement by lying on the ground was
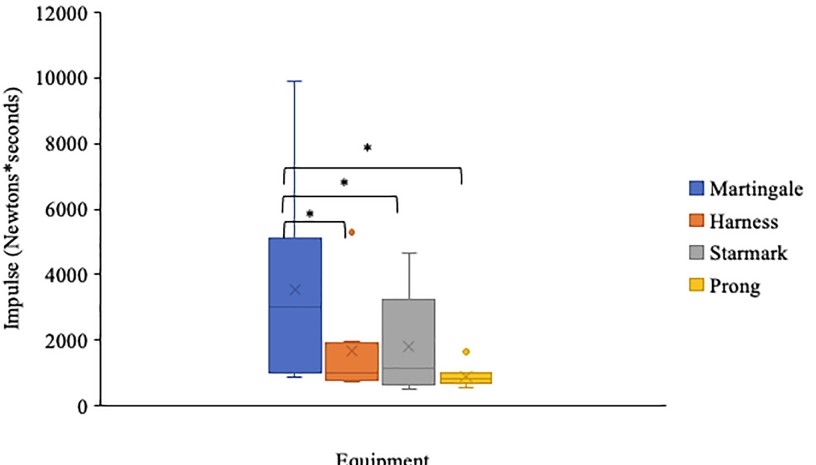

**Figure 3 Impulse over a 5-min walk for each equipment type and all dogs.** The box shows the interquartile range (IQR) from Quartile 1 to Quartile 3, Quartile 2, the median, is indicated by the horizontal line. Whiskers indicate the values within 1.5 times the IQR. Dots indicate outlier values larger than 1.5 times the IQR. "X" indicates the mean. Brackets with * indicate differences significant at $p < 0.008$.               

**Table 3 Total frequency of coded behaviors for all dogs for each 5-min walk.**

| Behavior | Martingale | Harness | Starmark | Prong |
|---|---|---|---|---|
| Whale eye | 0 | 0 | 0 | 0 |
| Look at handler | 80 | 100 | 69 | 87 |
| Bark | 2 | 3 | 0 | 5 |
| Yelp | 0 | 0 | 0 | 0 |
| Whine | 2 | 0 | 0 | 6 |
| Howl | 2 | 0 | 0 | 0 |
| Growl | 1 | 1 | 0 | 0 |
| Lip lick | 130 | 185 | 158 | 179 |
| Tucked tail | 1 | 1 | 0 | 1 |
| Crouched body | 0 | 0 | 0 | 1 |
| Yawn | 1 | 0 | 0 | 1 |
| Shake off | 1 | 0 | 0 | 0 |
| Quiver | 0 | 0 | 0 | 0 |
| Sniff | 245 | 189 | 197 | 170 |
| Balk | 4 | 6 | 8 | 16 |

only observed in one dog across all the equipment types. None of the dogs showed signs of distress or pain requiring termination of a walk.

Figure 4 shows the frequencies of the three analyzable behaviors for each equipment type. Friedman's tests showed no significant differences in frequencies of looking and sniffing among the four types of leash-equipment (looking at the handler: $\chi^2_{(2)} = 3.44$, $p = 0.33$; sniffing: $\chi^2_{(2)} = 4.47$, $p < 0.22$). Frequency of lip licking differed significantly

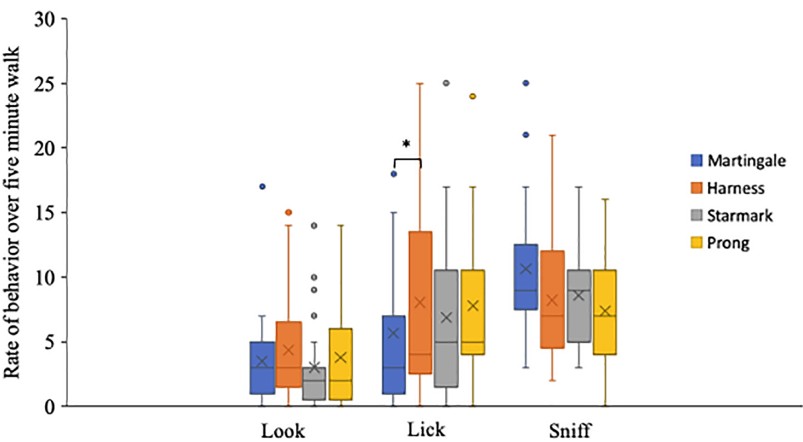

**Figure 4 Rates of looking, licking and sniffing across a 5-min period for all dogs.** Details for behavior rates in Table 3. The box shows the interquartile range (IQR) from Quartile 1 to Quartile 3, Quartile 2, the median, is indicated by the horizontal line. Whiskers indicate the values within 1.5 times the IQR. Dots indicate outlier values larger than 1.5 times the IQR. "X" indicates the mean. Brackets with * indicate differences significant at $p < 0.008$.

among the four types of leash-equipment (Friedman's test, $\chi^2_{(2)} = 8.17$, $p = 0.04$). Bonferroni-corrected ($p = 0.008$) *post-hoc* Wilcoxon signed-ranked tests found a significant difference in lip licking between the martingale and the harness ($Z = -2.65$, $p = 0.008$, r = 0.55).

## DISCUSSION

This is the first study to utilize a strain gauge to assess pulling force across four types of leash walking equipment. We were particularly interested to see if, as advocated by some dog trainers (*e.g.*, *Frawley, n.d.*; *Dogology, 2022*), the prong collar was superior at managing pulling behavior, but, whether due to its aversive design, there were any negative welfare effects.

The martingale was considered a baseline measure in this study as it is comparable to a flat neck collar that dogs are commonly walked on (*Dinwoodie, Zottola & Dodman, 2021a*; *Townsend et al., 2021*), but the tightening mechanism offered some safety if shelter dogs attempted to escape. As previous studies on flat collars have shown (*Carter, McNally & Roshier, 2020*; *Pauli et al., 2006*), dogs in our study pulled significantly more on the martingale than on any other equipment type. Although there were no statistically significant differences in pulling among the other three equipment types, in terms of trends, dogs pulled least on the prong collar and an intermediate amount on the Starmark collar and harness. Given that the prong collar was the most punitive equipment type in this sample, it is perhaps not surprisingly that the dogs experiencing potential aversive pressure from the metal prongs of the collar attempt to alleviate the pressure by reducing pulling.

*Shih et al. (2021)* found that dogs pulled more on a harness than a flat collar, however the harness they tested had a back-connection, whereas we used a front-connected harness. *Shih et al. (2021)* aimed to test claims that a back-connection harness encouraged pulling.

Because we were interested in equipment claimed to reduce rather than encourage pulling, we did not include a back-connection harnesses in this study.

Our finding that dogs pulled more on the martingale than on the other equipment types might be interpreted to imply that the dogs were experiencing more stress on the martingale because of the pressure that the collar would create on the neck (*Carter, McNally & Roshier, 2020*). However, stress-related behaviors were not increased on the martingale collar.

It is possible that dogs pulled more on the martingale precisely because it was more comfortable than the other leash types which are either more restrictive across the shoulders and chest (harness) or exert more aversive pressure on the throat (the Starmark and prong collars). If dogs were less constrained by the martingale, it is possible they were less stressed and thus, also lip-licked less. Lip licking has been proposed as an indicator of stress for dogs experiencing separation anxiety (*Palestrini et al., 2010*), trained with negative reinforcement (*Deldalle & Gaunet, 2014*), or experiencing a threatening situation (*Firnkes et al., 2017*). *Firnkes et al. (2017)* reported that lip licking was often paired with gaze aversion and thus might serve as an appeasement signal in conflict situations.

The significantly higher rate of lip licking we observed for the harness than the martingale is similar to *Grainger, Wills & Montrose*'s *(2016)* observation of (non-significantly) more lip licking on a harness than a flat collar. *Grainger, Wills & Montrose (2016)* compared stress-related behaviors between a flat collar and a back-connection harness in order to test whether, because of the forces collars impose on the neck (*Carter, McNally & Roshier, 2020*; *Pauli et al., 2006*), harnesses might be "better for canine welfare" (*Grainger, Wills & Montrose, 2016*, p. 62). However, because the difference in lip-licking rate they observed was not statistically significant, and, also in the opposite direction from their hypothesis, they concluded that it could not be determined whether welfare was compromised on either device. Similarly, in our data, it is also difficult to make any broad interpretations on any possible impaired welfare, or lack thereof, given that we found only one significant difference in one behavioral indicator.

We included sniffing in our ethogram as a possible indicator for stress, yet we did not see any significant differences in sniffing frequency. Like *Grainger, Wills & Montrose (2016)*, we found that both the collar and the harness yielded similar sniffing rates. While sniffing is often included as a metric for behavioral assessment of welfare, the literature does not offer consensus on whether it corresponds to poorer or better welfare. *Beerda et al. (1998)* noted sniffing as a sign of restlessness that was associated with aversive stimulation and this is often referenced in ethogram design (*e.g.*, *Grainger, Wills & Montrose, 2016*; *Shih et al., 2021*). However, other studies indicate that providing dogs with an opportunity to sniff may have a positive impact on welfare (*Binks et al., 2018*; *Tod, Brander & Waran, 2005*). In fact, *Petrean et al. (2023)* included sniffing —operationally defined as "the nose held close to or touching a surface and/or sniffing the surface" (*Petrean et al., 2023*, p. 3) as an indicator of a negative state and exploring the environment, defined as "walks with nose close to surfaces or sniffing objects" (p. 4) as an indicator of positive emotional state when assessing stress in shelter dogs. Dog trainers

commonly refer to sniffing as a displacement behavior used to diffuse stress or tension (*e.g.*, *Knowles, 2017*; *McConnell, 2016*). Given that, similarly to other recent studies (*e.g.*, *Grainger, Wills & Montrose, 2016*; *Petrean et al., 2023*; *Shih et al., 2021*), we did not find any association between sniffing frequency and any other behavioral indicators, more research is warranted to assess sniffing after positive or negative situations to understand the underlying emotions that may drive this behavior.

Looking at the handler was included in the ethogram because prior studies (*Grainger, Wills & Montrose, 2016*, *Ogburn et al., 1998*; *Shih et al., 2021*) have suggested it as an indicator of stress. However, like sniffing, gaze has been subject to multiple interpretations, including as a sign of seeking reassurance (*Wanser & Udell, 2019*) and as a form of referential looking (*Shih et al., 2021*). In our data we found no statistically significant differences between the leash equipment types in looking to the handler, however, there was a trend for the frequency of looking to the handler to be highest when walked on the harness, in line with previous research. *Shih et al. (2021)* found that dogs wearing a harness looked more to the experimenter during a food-reward condition than dogs wearing a collar. *Shih et al. (2021)* proposed that looking was more frequent when wearing a harness because dogs were less inhibited in movement than when wearing the neck collar. This would also account for the trends in our data.

It is important to note that while aversive collars are critiqued by welfare organizations (*American Veterinary Society of Animal Behavior (AVSAB), 2021*; *RSPCA, 2024*) and prong collars are banned in several countries (*Makowska & Cavalli, 2023*) because of their aversive design, any type of leash walking equipment has the potential to be aversive for its wearer. Harnesses put pressure across the chest and back and can restrict shoulder rotation and gait (*Lafuente, Provis & Schmalz, 2019*; *Pálya et al., 2022*). For a dog that is sensitive around its joints or limbs, a restrictive harness might be more aversive than a collar that places pressure solely on its neck (*Grainger, Wills & Montrose, 2016*; *Pauli et al., 2006*).

## Limitations and future directions

One limitation of this study is order effects. We walked dogs on the equipment types in increasing order of putative severity out of concern that the dogs, if first walked on more aversive equipment, might have experienced discomfort that could influence their behavior on future walks with the same handler. This, however, leaves open the possibility, that any changes in behavior across the four walks could have been due to growing exhaustion across the walks rather than the equipment. We believe this is unlikely to be a factor because we gave the dogs breaks of at least 2 h between each walk and kept the walks to 5 min. The four walks each dog experienced in a day is a total of just 20 min of activity. Informally, all dogs appeared eager to come out of their kennels and interact at the start of each walk. Although all dogs were walked on all forms of equipment in the same order, the first walk of each day occurred at different times of day and, thus, circadian effects were unlikely.

The dogs' behavior in general and pulling in particular could have been impacted by increased familiarity with the handler and the walk route. Insofar as this could be a concern

it mirrors the experience of a pet dog going on regular neighborhood with its owner: Owners still report concerns with leash pulling on their daily walks (*Blackwell et al., 2008*; *Townsend et al., 2021*).

The walks the dogs in this study received were likely shorter than a typical pet dog's walk with its owner, but, after piloting 10-min walks with some dogs, we felt that 5 min was an appropriate duration to obtain a complete assessment of all the behavior a dog might express on a walk and reduced the risk of tiring the dog and inconveniencing the shelters where the study was carried out. It is reasonable to assume that any pulling and other reactions to the leash equipment were more likely to occur early in a walk when the dog is most excited and experiencing the equipment for the first time.

Another limitation, and avenue for future research, lies in the nature of the walks the dogs participated in. The dogs in our study were walked with minimal interference from the handler in order to provide the best foundation to assess impact of equipment design on dog-initiated pulling. However, most dog walkers are probably more interventionist than we were. Further studies could investigate the impacts of different kinds of leash equipment on more typical dog walks with more interference and additional pulling tension imposed by the human dog walker.

Finally, we note that with a larger sample size, it might have been possible to obtain more clarity on the presently borderline differences in sniffing between equipment types.

## CONCLUSIONS

When trying to mitigate problematic leash pulling, dog owners can select from a wide variety of leash equipment types. These tools vary, particularly, in the degree of aversive pressure applied to punish pulling. We were particularly interested to see whether the prong collar was any better at mitigating leash pulling than other commonly used types of leash equipment, and whether there were any welfare impacts effects to its use. We were able to demonstrate using a strain gauge that dogs pulled most on a martingale collar compared to the other leash equipment types. There were no statistically significant differences in pulling between the putatively most aversive tool, the prong collar, and the other less aversive types. Behavior analysis did not provide clear evidence that any of the four equipment types resulted in impaired welfare. Although these results might indicate to some that dogs are not stressed while on tools designed to create some aversive pressure like the Starmark or prong collar, we caution generalizing these results to the wider pet dog population. Dogs in this study were walked without any additional tension or pressure from the handler in a relatively controlled environment. These conditions likely do not hold on many typical dog walks. Further research should investigate behavioral and welfare impacts across different equipment types when dogs are walked by a variety of handlers in other walking scenarios.

## ACKNOWLEDGEMENTS

We would like to thank the staff and volunteers at Yolo County Animal Services and Elk Grove Animal Services. Without their willingness and support, this study would not have

been possible. Additional thank you to the dedicated undergraduate reviewers for coding the videos that were crucial for analysis.

### Funding
The authors received no funding for this work.

### Competing Interests
The authors declare that they have no competing interests.

### Author Contributions
- Anamarie C. Johnson conceived and designed the experiments, performed the experiments, analyzed the data, prepared figures and/or tables, authored or reviewed drafts of the article, and approved the final draft.
- Clive D. L. Wynne conceived and designed the experiments, authored or reviewed drafts of the article, and approved the final draft.

### Animal Ethics
The following information was supplied relating to ethical approvals (*i.e.*, approving body and any reference numbers):

This study was approved by the Institutional Animal Care and Use Committee (IACUC) of Arizona State University (Protocol Number: 22-1934R).

### Data Availability
The raw data is available in the Supplemental Files.

### Supplemental Information
Supplemental information for this article can be found online at http://dx.doi.org/10.7717/peerj.18131#supplemental-information.

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
