# Peer review of "Comparing efficacy in reducing pulling and welfare impacts of four types of leash walking equipment"

_PeerJ, doi:10.7717/peerj.18131_

## Round 0.1 · original submission · Major Revisions

I was fortunate to receive two excellent reviews from experts who outlined a number of points in need of consideration as you revise your work. You attempted to address the issue of order effects in terms of exhaustion but please also note the issue of familiarity as raised by Reviewer 2. Given that you did not know a priori which type of collar may impact the dogs the most, it seems it might have been preferable to counterbalance the order. You could have also walked the dogs on different days to mitigate against carry-over from the same handler. Please take care to be more cautious in your interpretations in light of these possible effects as well as the lack of knowledge about prior leash experience in shelter dogs. It would be helpful to include images of the different types of collars.
Although the paper is technically well-written, the introduction and discussion are lacking a coherent organization. Please ensure that paragraphs are fully developed with each containing an introductory statement, supporting body and conclusion. Sometimes it is unclear what the key point is from the list of facts provided. More structuring of the information would be helpful. It would be useful to indicate the types of collars and their use, the problem behaviors that precipitate or are exacerbated by use of the different collars. It was unclear until the end of the introduction whether the aim was to address effectiveness or welfare related to the collars.
Is a 5-min walk sufficient to detect differences with different collars? In terms of ecological validity, it seems short for representing a typical walk.
Why does line 260 begin with "But?"
You need to provide a clear rationale for the behaviors selected for coding and indicate to the reader what these behaviors indicate. Please include clear predictions/hypotheses.
Was the strain gauge validated prior to testing?
Please avoid using "Since" in non-temporal context.

·

Basic reporting

Authors provide thorough background information with appropriate citation. Figures and tables are clear. It would be good if authors could add the research hypotheses at the end of the introduction, and in the discussion, discuss whether the hypotheses were supported by this study. Overall, the article is well written using unambiguous English. Authors are encouraged to review the article again to improve the linking between paragraphs and the overall flow, especially the discussion part. Other minor comments as below.

Line 169: …was guided and provided “by” shelter staff…

Experimental design

The research questions were well defined and the research design and results were relevant. Authors are encouraged to provide more details about the methodology (see below comments). Finally, although authors mentioned the order effect in the limitation, it would be much appreciated if more justification could be provided as the order effect could potentially significantly affect the result.


Line 173-175: Please define “temperamentally suited” and explain how is it related to the study and subject selection. Also, were these shelter dogs regularly walked? If yes, what equipment was used?

Line 177-178: Please provide some background information of each shelter. Any difference in terms of setting, environment, practice, animal characteristic and staff/vlounteer, etc.?

Line 181-186: To make if more reader friendly, it would be good if authors could briefly introduce the design and mechanism of each equipment. Also, why were these types of equipment chosen for this study?

Line 188-197: How was the pulling force data recorded? Was it recored in a real time manner and could later be exported?

Line 211-212: How did owner determine “aversiveness”? Not sure why front-connection harness was more aversive than martingale collar.

Line 211-215: Since all walks were completed in the same order of equipment, would there be an order effect on your result? For instance, dogs were tired/less active by the end of the last equipment and thus tended to pull less on it. Also, were all walks for each dog completed on the same day? If yes, then it would take at least 6 hours to complete all walks. Similarly, the dog might be less active by the end of the day, or most active around the meal time. Such order could potentially affect the result. Looking at Fig 2., it does have a trend that dogs pulled harder on the first tested equipment (martingale) and least on the last tested one (prong).

Line 217-229: Were all walks completed by the same person? If not, the handling, entire procedure and dogs’ response were less consistent which might greatly affect the result.

Line 251: Not sure about the formula here: total force (Newtons) x seconds. The force was not consistent over the time, so wouldn’t it be the “integral of the force-time”? Again, this goes by to my previous question. How was the pulling force data recorded? Was it recored in a real time manner and could later be exported?

Line 302-320: As authors mentioned, there was only one behavior that was significantly different. Also, looking at Fig 3-licking behavior, the median of harness was although higher than martingale but lower than starmark and prong, and harness had a very wide IQR compared to others. Therefore, one should be aware not to over interpret the result.

Line 352-354: “Dogs experiencing positive…negative reinforcement” Not sure why authors mentioned this. Please clarify.

Validity of the findings

This is the first study to assess dogs' pulling force and behavior on four types of equipment using a strain gauge. This is an important and novel study that adds to our understanding about different leash walking equipment. However, regarding the behavioral result, as previously described, authors are encouraged not to over-interpret the result.

·

Basic reporting

Thank you for submitting this paper to PeerJ. It is a good addition to this area of work.
Overall the paper is clear and professionally written. The introduction provides a good foundation/ literature review for the study.
The only point I would make here is that when a previous study is introduced and discussed, and it is obvious that various sentences refer to the same study, the reference does not need to be repeated. For example, in the paragraph lines 88-101, Ogburn et al., 1998 does not need to br repeated so often as it is obvious you are still discussing their work. Similarly with Grainger et., 2016 does not have to be in there twice. The reference only needs to be repeated if you feel there is some doubt about the source.
This is true also for paragraph 103-107. Shih et al., 2020 does not need to be repeated.
Please carefully examine each paragraph and only repeat a reference where necessary.

Experimental design

The approach to the study appears to be generally appropriate. Here are a few issues I think should be addressed.
Line 173: It appears to be a big assumption that shelter dogs would have minimal prior exposure to leash equipment. Animals enter shelters from various sources including previously owned dogs which may have been regularly walked on leashes. Also, while in the shelter, the dogs were presumable walked daily by volunteers or staff and depending on their length of stay may have been very familiar with the equipment. The best that can be said is that the digs had an unknown history of leash walking before entering the shelter.
Line 174-5: It is not clear how shelter staff helped choose dogs and you used random number at the same time. Also, what does temperamentally suited mean? And as an aside, how would shelter staff know this without having walked them? Were there other exclusion criteria such as size, age, health status, etc.
Leash equipment: Many readers are not going to know what a Starmark collar is and how it differs from a prong collar. Please explain. Also, clarify that the Starmark and prong collars were fitted to self release when the dog was not pulling. In addition, add that the martingale tightens as the dog pulls so not the same as a ‘normal ‘ dog collar.
Strain gauge: Please describe how this was calibrated and checked for accuracy.
Video recording of walk: I am having trouble seeing how it was possible to hold the video camera on a stick with the other hand while walking shelter dogs (which, as you state, may have not been familiar with leash walking). Also, how is this 360 degree filming?
Procedure: While the order of the walks following least aversive to most aversive is noted as a potential limitation to the study, I’d be interested to understand why you chose to do it this way and not with a random order. Please add how long the handler and dog remained in the yard before exiting and starting the recording. Was the dog excited or scared when encountering the handler initially or every time?
Line 260: Data is a plural word

Validity of the findings

Please also add in the results whether there were some video footage not recorded because of mistakes or dog behaviour. Also, were there any signs of distress as mentioned in 231 observed?
Discussion
Line 295: Is the martingale a flat collar, not like a flat collar?
Line 302: Is this Shih et al., 2021 a or b? Please check all Shih et al., 2021 and identify if a or b.
Paragraph 322-330: It seems important to mention that this study found a difference to Shih et al., 2021(a) in terms of on which leash the dogs pulled the most. The previous study found dogs pulled more in a harness where your study found the opposite to be true. Can you suggest an explanation?
I believe the paper should have a longer discussion about the pulling impulse across the collar types. For example, the least pulling was with the prong collar.
Line 328: Please check the English in this last sentence.
Line 373: You mention exhaustion as a possible reason for the results but have you also considered increased familiarity with the handler and the walk site?
Conclusion
In Line 157 you set out the aim of the study. Please ensure the conclusion states whether you met this aim. You sort of address this in the conclusion but I feel I need more. For example, why were you testing the prong collar at all if you think it is aversive. Maybe you would never recommend it. You are completely correct in saying this study did not really assess the welfare implications of the prong collar.

Additional comments

My only general comment is alluded to earlier; Why do research into a collar that most people believe is aversive?

---

## Round 0.2 · Minor Revisions

Thank you for a thorough revision. I was able to have two previous reviewers review the revision. One reviewer thinks the paper is now publishable but the other cautions you to provide even further clarification on aversiveness and the potential confound of walking order. Around line 63, can you please describe why a prong collar is aversive for those readers that are not familiar.

Please report the results of your reliability analyses for each behavior that was coded and analyzed.

Lines 424-429 be clear here whether you take looking to the handler to be positive or negative.

I just have a few extremely minor grammatical corrections for you to make before final acceptance:

Line 26, either change “which” to “that” or add a , in front of “which.”
Line 55, add , before which. Please check throughout for similar instances where a , is needed.
Lines 39, 151, 364, 424, 482, change “While” to “whereas” or “although..”
Line 189, place a , after study
Again, please avoid single sentence paragraphs (e.g., lines 182-185; 256-258).
Line 268, After “body” please insert “; however, the…”
Line 321 data “were…”
Line 407, change “were similar in” to “yielded”

·

Basic reporting

The paper is well written using unambiguous and professional English. Appropriate background information and references were provided. Figures and tables are clear. Minor comments as below.

Line 265-271: Thanks for the explanations about “aversiveness”. If I understand it correctly, the “aversiveness” here indicated how dogs felt about the equipment rather than the welfare consequence of the device. For instance, dogs generally preferred flat collars over harnesses so harnesses were “more aversive”. It is a little bit confusing because “aversive” is generally used to describe the welfare consequence of a particular thing, such as the aversive training technique. In this case, it is generally believed that harnesses are better for dogs’ welfare and health, so it is a bit confusing to say front-connection harness is more aversive than martingale collar. Please clarify it.

Experimental design

The experiment is overall well designed. Authors provided sufficient details about the study. My main concern is the walking order (see below).

Regarding the walking order (lines 260-272 and lines 440-448), authors mentioned that all walks were completed in the same order of increasing potential level of aversiveness, and that dogs were unlikely to be affected by growing exhaustion across the walks. I am not sure about the arguments. First of all, is there an objective method that justifies the level of aversiveness? Also, shelter dogs are generally more aroused (more barking and higher cortisol level) in the morning when they first met people or being fed (see references below). Therefore, it was very likely that dogs tended to pull harder on the equipment tested first in the morning. In this study, martingale collar was tested first and dogs also pulled most on a martingale collar, which is very likely to be caused by the high arousal state of dogs in the morning rather than the type of collar.

Fernandez, Eduardo J., Wes Anderson, and Amanda Kowalski. "Evaluation of an automated response‐independent schedule on the behavioral welfare of shelter dogs." Journal of the Experimental Analysis of Behavior 120.1 (2023): 50-61.
Sales, G., et al. "Noise in dog kennelling: is barking a welfare problem for dogs?." Applied Animal Behaviour Science 52.3-4 (1997): 321-329.
Beerda, Bonne, et al. "Chronic stress in dogs subjected to social and spatial restriction. II. Hormonal and immunological responses." Physiology & behavior 66.2 (1999): 243-254.

Validity of the findings

Conclusions: Authors indicated that dogs pulled most on a martingale. However, the device used in this study did not differentiate the direction of the pulling (dog vs handler). Would it be possible that handlers tend to pull more or less on certain equipment? For instance, the higher impulse of martingale collar was because the handler instead of the dog tended to pull harder on the leash.

·

Basic reporting

no comment

Experimental design

no comment

Validity of the findings

no comment

Additional comments

Thank you for your revision of the manuscript and for addressing everything I raised. There is still at least one place where 'data' is treated as a singular but that should be picked up in the editing process.

---

## Round 0.3 · accepted · Accept

Thank you for addressing these final minor issues.